# A Novel Biosensor and Algorithm to Predict Vitamin D Status by Measuring Skin Impedance

**DOI:** 10.3390/s21238118

**Published:** 2021-12-04

**Authors:** Jin-Chul Heo, Doyoon Kim, Hyunsoo An, Chang-Sik Son, Sangwoo Cho, Jong-Ha Lee

**Affiliations:** 1Department of Biomedical Engineering, School of Medicine, Keimyung University, Daegu 42601, Korea; washingbuffer@gmail.com; 2Samsung Research, Samsung Electronics, Suwon 16677, Korea; doyoon07.kim@samsung.com (D.K.); hyun-soo.an@samsung.com (H.A.); 3Division of Intelligent Robot, Daegu Gyeongbuk Institute of Science and Technology, Daegu 42988, Korea; changsikson@dgist.ac.kr; 4The Center for Advanced Technology in Testing Human Factors, Keimyung University, Daegu 42601, Korea; chosw@kmu.ac.kr

**Keywords:** biosignal, impedance, InBody, multiple machine learning, vitamin D

## Abstract

The deficiency and excess of vitamin D cause various diseases, necessitating continuous management; but it is not easy to accurately measure the serum vitamin D level in the body using a non-invasive method. The aim of this study is to investigate the correlation between vitamin D levels, body information obtained by an InBody scan, and blood parameters obtained during health checkups, to determine the optimum frequency of vitamin D quantification in the skin and to propose a vitamin D measurement method based on impedance. We assessed body composition, arm impedance, and blood vitamin D concentrations to determine the correlation between each element using multiple machine learning analyses and an algorithm which predicted the concentration of vitamin D in the body using the impedance value developed. Body fat percentage obtained from the InBody device and blood parameters albumin and lactate dehydrogenase correlated with vitamin D level. An impedance measurement frequency of 21.1 Hz was reflected in the blood vitamin D concentration at optimum levels, and a confidence level of about 75% for vitamin D in the body was confirmed. These data demonstrate that the concentration of vitamin D in the body can be predicted using impedance measurement values. This method can be used for predicting and monitoring vitamin D-related diseases and may be incorporated in wearable health measurement devices.

## 1. Introduction

Vitamin D (25(OH)D) is an essential nutrient for good health. Insufficient vitamin D levels cause rickets in children and osteomalacia, bone pain, and muscle weakness in adults [1]. In addition, Vitamin D plays an essential role in the musculoskeletal system and has recently been reported to influence chronic disease conditions such as cancer, obesity, metabolic syndrome, diabetes, and cardiovascular disease [2,3]. On the other hand, excessive levels of vitamin D in the blood can cause nausea, vomiting, muscle weakness, pain, loss of appetite, dehydration, excessive urination, thirst, and kidney stones. It can also cause kidney failure, irregular heartbeat, and even death [4,5].

In the present day, many people have lower vitamin D levels than recommended, and vitamin D deficiency is rampant. Measuring serum vitamin D level is expensive, and universal testing is not available [6]. Quantification of vitamin D can be performed by competitive protein binding (CPB) assays, radioimmunoassay (RIA), chemiluminescence immunoassay (CLIA), liquid chromatography with UV detection (LC), and liquid chromatography–mass spectrometry. These methods are labor-intensive and technically complex [7,8]. Moreover, most of these methods use blood samples and put considerable pressure on users. A non-invasive method of measuring vitamin D using hair sample exists, but the analysis uses classical methods, such as HPLC [9]. Other conditions that predict vitamin D levels include normal blood levels of calcium and phosphorus, high or high levels of parathyroid hormone, normal to high levels of total alkaline phosphatase, urinary calcium excretion rates, and vitamin D binding protein analysis [6,10,11].

A blood test is an accurate way to measure the amount of vitamin D in the body. Methods for indirectly measuring vitamin D include hair analysis [9] and Insufficiency Prediction score reflects individual characteristics using a multivariate logistic regression model for overweight, physical activity, winter season, sun exposure, etc. [12]. The amount of vitamin D in mouse skin was measured using an impedance-based quantum analyzer as a non-invasive method using biosignals [13]. There is a report that body mass index, body weight, and body fat are related to the amount of vitamin D in the body [14,15]. However, there are no studies that have predicted levels of vitamin D by measuring skin impedance in humans.

As a branch of artificial intelligence, machine learning has made remarkable progress. Deep learning methods such as recurrent neural network (RNN) and convolutional neural network (CNN) are emphasized [16]. Machine Learning has allowed for enhancement in analytical capabilities of these various biosensing modalities. Biosensor data can be analyzed from various perspectives using fusion data. The connection between machine learning and biosensors will expand significantly as tools for detection, analysis, and diagnostics [17].

Recently, following the development of smart phones and wearable devices, a method facilitating easy assessment of vitamin D level was developed. It utilizes a smartphone accessory, a mobile app, and a test strip that enables the colorimetric detection of vitamin D using a novel gold nanoparticle-based immunoassay [18], developed through lab-on-a-chip technology. This analytical method, however, does not replace the existing blood analysis. Along similar lines, an easier and more efficient non-invasive analytical method based on a simple score (a vitamin D deficiency predictive score) to identify adults at risk for vitamin D deficiency could not replace blood tests either [12]. This method has reliability concerns with respect to quantification, since its assessment of the user’s behavior and body characteristics using a questionnaire simply informs the level of risk involved. Therefore, a scientifically quantifiable and non-invasive method is necessary for measuring vitamin D levels. There are studies that have confirmed the relationship between vitamin D and impedance using mice [13], but there are no studies on humans.

The present study proposes a novel method for measuring vitamin D levels through bioelectrical signal measurement using impedance measurements. In this method, the body characteristics of study subjects were analyzed, and their correlation with vitamin D was assessed. Subsequently, the optimum frequency at which vitamin D can be measured was determined, and an algorithm predicting vitamin D level based on impedance measurement was developed. Thus, in this study, we present a simple method for measuring vitamin D based on skin impedance measurement.

## 2. Materials and Methods

### 2.1. Experimental Design

In this study, user information, body composition measurement obtained using an InBody scan [19], arm impedance measurement, and the vitamin D concentration in the blood were analyzed. The measured values were investigated for correlation between each element using multiple machine learning analyses, and an algorithm predicting the concentration of vitamin D in the body using the impedance value was developed (Figure 1).

### 2.2. Participants

A total of 26 participants, with ages ranging from 20 to 60 years, living in Daegu City (metropolitan city) during 2020, were included in the study. For body composition measurements, obesity level, body composition, and skeletal muscle fat were measured using InBody 770 (InBody, Seoul, Korea) equipment.

### 2.3. User InBody Measurement

Bioelectric impedance analysis was performed to determine body composition. InBody 770 (the 4-pole 8-point touch-type electrode system, direct multi-frequency measurement method, and simultaneous multi-frequency impedance measurement method were used) was used to analyze the body composition. Participants performed InBody measurements on the same day as the health checkup, and the average of the three repeated measurements was used for the analysis. In subsequent analyses, the data from the measured values were processed. Values pertaining to obesity score, body composition, skeletal muscle/fat, muscle by part, extracellular water ratio, and body fat by part collected from the InBody device were analyzed for their correlation with vitamin D level. Data related to parameters that could be used to predict vitamin D level were collected from the participants. The data collected (32 types) included basic information (age and sex), obesity analysis (body mass index and body fat percentage), body composition (body water, protein, minerals, body fat, muscle mass, and lean body mass), skeletal muscle and fat analysis (skeletal muscle mass), muscle analysis by region (right arm (kg, %), left arm (kg), trunk (kg), right leg (kg), and left leg (kg)), extracellular water ratio, body fat analysis by region (right arm, left arm, trunk, right leg, and left leg), and other parameters (intracellular water, extracellular water, basal metabolic rate, abdominal fat percentage, and body cell mass). Details related to the overall data collected from the participants are presented in Appendix A.

### 2.4. Participant Health Checkup

The study was performed at Keimyung University Dongsan Medical Center, and health checkup was performed based on the results from the medical examination of the participants. Health checkup parameters used in the analysis included the participant’s basic information (age and gender), physical measurements (waist circumference and body mass index), blood pressure (systolic blood pressure, diastolic blood pressure, and pulse), general blood tests (hemoglobin, packed cell volume, white blood cells, red blood cells, platelets, mean corpuscular hemoglobin, red cell distribution width, platelet distribution width, erythrocyte sedimentation rate, and mean platelet volume), leukocyte levels (neutrophils, lymphocytes, monocytes, eosinophils, and basophils), lipid levels (total cholesterol, HDL-cholesterol, and LDL-cholesterol), triglyceride level, liver function tests (aspartate aminotransferase, alanine aminotransferase, gamma-glutamyl transferase, albumin, total protein, total bilirubin, direct bilirubin, alkaline phosphatase (ALP), lactate dehydrogenase (LDH), and albumin/globulin ratio), glucose tests (blood sugar, HbA1c-NGSP, insulin, and homeostasis model assessment ratio), tests related to kidneys and pancreas, and other tests (blood urea nitrogen, uric acid, sodium, potassium, phosphorus, creatinine, amylase, calcium, GFR (CKD-EPI), and glomerular filtration rate), as well as vitamin D levels. Details on the health examination data obtained from the participants are presented in Appendix A.

### 2.5. Impedance Measurement

For the skin impedance measurement, bioimpedance measurements from the forearm were obtained through a frequency (20 Hz–1 MHz) scan using an impedance measuring instrument (E4980AL, Keysight). Impedance was measured using 400 frequencies to scan 200 target points, while maintaining a constant interval in the 100 Hz–1 MHz frequency range and 200 frequency points in a certain range of the log-converted frequency as the measurement frequency.

### 2.6. Correlation between InBody Measurements and Vitamin D Levels in the Body

The algorithms used for data analysis were Statsmodels (version 0.12.2) on Python 3 and the linear regression model and Scikit-Learn (version 0.22.2) on Python 3, a 10-fold cross-validation and the root mean for the performance evaluation methods and the squared error (RMSE) were used. Matplotlib (version 3.2.1) and Seaborn (version 0.10.1) were used as visualization tools. For data analysis elements, data collected from the participant’s gender, age, and measurement device were used, and the correlation with vitamin D was analyzed, and measurement parameters that could be used to estimate the amount of vitamin D were selected. In this study, data from 26 people were used for analysis, and in order to improve the performance evaluation of the dataset, cross-validation and error indicators were converted into units similar to the actual values for analysis. The analysis values were used as statistical graphics, using a visualization library. The test for normality is performed by investigating whether an independent variable (i.e., factor) with continuous values follows a normal distribution (Shapiro–Wilk test). Factors that did not satisfy normality (*p* > 0.05 identify significant differences with probabilities for test statistic) were retested after log transformation, and those that did not satisfy normality after retesting were excluded from the analysis. Among the 31 variables (gender excluded), 19 variables were found to satisfy normality (Appendix A).

The input variable was the gender dummy variable (reference, male), the variable considered after the normality test was used, and the vitamin D value was used as the output variable. When designing the model, the variable (i.e., factor) selection method did not have sufficient sample size to construct a regression analysis model as a stepwise selection method, and so the significance levels were adjusted to 0.5 and 0.55, respectively, when selecting and removing stepwise factors. Due to the possibility of collinearity or multicollinearity problems if the significance level was higher than the usual standard (0.05, 0.1) when a factor was selected and removed, the final factor was determined through a post-processing process in which factors with a relatively high significance level were removed step by step, in case of a multicollinearity problem.

### 2.7. Relationship between Health Checkup and Vitamin D Levels in the Body

To determine the relationship between health checkup and Vitamin D, the analysis was performed in the same way as in Materials and Methods 2.6 (Section 2.6). Five of the parameters analyzed were found to be statistically insignificant after retesting for normality. The parameters excluded were general blood test (hemoglobin and erythrocyte sedimentation rate), glucose test, blood sugar, glycated hemoglobin (HbA1c)-NGSP, and potassium.

The model designed included a dummy variable of gender (reference, male) as the input variable, 45 analysis values considered after the normality test, and the vitamin D value as the output variable. When designing the model, a stepwise selection method was used to select the variables (factors). To construct a regression analysis model, the significance levels of stepwise factor selection and removal were adjusted to 0.05 and 0.1, respectively.

### 2.8. Relationship between Skin Impedance and Vitamin D Levels in the Body

Impedance measurements were obtained from the forearm, and the variables used for the analysis were the participants’ basic information (age and gender) and the participants’ impedance measurement values. The analysis algorithm was performed in the same way as in Materials and Methods 2.6 (Section 2.6). When designing the model, a stepwise selection method was used as the criterion for selecting the variables. When a factor was selected and removed, there was the possibility of collinearity or multicollinearity problems occurring if the significance level was higher than the usual standard (0.05, 0.1). Therefore, when a multicollinearity problem occurred, the final factor was determined through a post-processing process in which the factor with a relatively high significance level was removed step by step. A predictable vitamin D measurement frequency was selected through the normality test and the model design, and a skin-impedance-based vitamin D prediction algorithm was developed by analyzing the correlation between the participant’s impedance measurement value and the blood vitamin D concentration.

## 3. Results

The correlation with vitamin D in the body was analyzed for parameters such as sex, age, obesity, body composition, skeletal muscle/fat, muscle by part, and the extracellular water ratio collected from the InBody device, and the measurement parameters that could be used to estimate the amount of vitamin D were selected.

### 3.1. Correlation between InBody Measurements and Vitamin D Levels in the Body

Upon data analysis, the F statistic, the ratio of two quantities that are expected to be equal under the null hypothesis, of the regression model showed a level of 2.803 at *p* (probability value) = 0.0814, R^2^ = 0.196 for the regression formula which showed an explanatory power of 19.6%, AIC (Akaike Information Criterion) = 27.91, and BIC (Bayesian Information Criterion) = 31.68 (Table 1). Analysis of the relationship between the factors collected from the InBody device and the vitamin D level revealed that the right leg and the body fat percentage had an effect on muscle analysis by region. In particular, body fat percentage (t = 2.355, *p* < 0.05) was found to have a statistically significant effect (Table 2). To confirm the linearity of the model, the predicted values ($\hat{y}$) and the residuals (y − $\hat{y}$) were compared (Figure 2A). The predicted value of vitamin D (log value) showed a larger error corresponding to the values less than 3.1 and 3.65 or more. Using a QQ plot (Quantile-Quantile Plot), it was reviewed whether the assumption of normal distribution was appropriate. Tests for normality of errors using a QQ plot, performed to confirm the normality of the residuals, showed that the normality was satisfied at the significance level of 5% with *p* = 0.6226 (Figure 2B). Statistical cross-validation was performed by dividing all of the data into sub-data for training and validation (training: 80%, validation: 20%). The mean RMSE and standard deviation were estimated for 100 trials using random seeds and were found to be 0.4 ± 0.1072. (Figure 2C).

### 3.2. Relationship between Health Checkup and Vitamin D Levels in the Body

The correlation with vitamin D in the body was analyzed for 50 parameters such as physical measurement, blood pressure, general blood test, white blood cell test, lipid levels, liver function test, glucose test, and kidney/pancreas/other tests in the participants’ health checkup parameters to select the measurement parameters that could be used to estimate vitamin D levels. Upon data analysis, the F statistic of the regression model showed a level of 9.981 at *p* = 6.64 × 10^−5^ and R^2^ = 0.714 for the regression formula which showed an explanatory power of 71.4% (Table 3). Following the analysis of the association between the health checkup parameters and vitamin D, parameters such as female (t = 4.447, *p* < 0.005), basophils (t = −4.765, *p* < 0.005), albumin (t = 4.116, *p* < 0.005), modification of diet in renal disease (MDRD) (t = −2.38, *p* < 0.05), and LDH (t = 2.239, *p* < 0.05) were investigated, and the statistically significant effect was confirmed (Table 4). Upon comparison of the predicted values ($\hat{y}$) and the residuals (y − ($\hat{y}$)) to check the linearity of the model, the predicted value of vitamin D (value) showed a larger error below 3.38 (Figure 3A). Tests for normality of errors using a QQ plot, performed to confirm the normality of the residuals, showed that the normality was satisfied at a significance level of 5% with *p* = 0.6646 (Figure 3B). Statistical cross-validation was performed by dividing all of the data into sub-data for training and validation (training: 80%, validation: 20%), and using random seeds, the average RMSE and standard deviation were estimated to be 0.2896 ± 0.0807 (Figure 3C).

### 3.3. Model to Analyze Vitamin D Level Based on Skin Impedance Measurements

The purpose of this study was to analyze the correlation of bioimpedance with the vitamin D concentration in the body by observing the change in the skin impedance of the forearm measured using an impedance device, and to identify the impedance measurement frequency that could be used to predict vitamin D levels. From the normality tests, three impedance frequencies (21.1 Hz, 22.2 Hz, and 23.4 Hz) were found to satisfy normality. In the model design, the three frequencies identified from the normality tests were set as the input variables and the logarithmic value of vitamin D was set as the output variable. Variable selection criteria were analyzed by defining the significance levels to be 0.05 and 0.1, respectively, when selecting and removing factors.

Upon data analysis, the F statistic of the regression model showed a level of 4.543 at *p* = 0.0435, R^2^ = 0.159 for the regression formula, which showed an explanatory power of 15.9% (Table 5). Analysis of the correlation between the collected frequency information and the vitamin D value confirmed that the impedance measurement frequency of 21.1 Hz (t = 2.131, *p* < 0.05) appeared to have a statistically significant effect (Table 6). From the comparison of the predicted values ($\hat{y}$) and the residuals (y − $\hat{y}$) to confirm the linearity of the model, it was observed that the predicted vitamin D value at 21.1 Hz showed a large error trend at values less than 3.15, between 3.2 and 3.5, and above 3.65 (Figure 4A). Tests for normality of errors using a QQ plot, performed to confirm the normality of the residuals, showed that the normality was satisfied at a significance level of 5% with *p* = 0.6285 (Figure 4B). For statistical cross-validation, the data were divided into sub-data for training and validation (training: 80%, validation: 20%), and the mean RMSE and the standard deviation during 100 validations using random seeds were estimated to be 0.3789 ± 0.1183 (Figure 4C).

Using the selected frequency of 21.1 Hz, the correlation between the impedance values and the vitamin D level confirmed through blood analysis was analyzed using the linear regression equation. A value of R^2^ = 0.7547 (y = 0.002x − 11.086) was obtained for the regression equation, showing 75% of the explanatory power (Figure 5).

## 4. Discussion

The recent advancements in the technology of wearable devices and an increased interest in health have resulted in easy and convenient diagnostic systems. Consequently, health assessment can be performed by analyzing various biosignals. However, quantification of vitamin D has so far been predominantly performed through blood analysis. This study aimed to develop an algorithm that can predict vitamin D levels using impedance measurements rather than the existing blood measurement method.

We tried to develop an algorithm to predict the concentration of vitamin D in the body based on skin impedance. In order to select a frequency at which impedance can be measured, the 20 Hz–1 MHz frequency band was divided into 400 sections on the participant’s skin to check the impedance. The data obtained from InBody confirmed the correlation between body composition content and skin impedance through linkage analysis. To predict the level of vitamin D in the body using impedance, the frequency band of 21.1 Hz was confirmed through statistical analysis using the measured impedance value. As a result of confirming the predictive value for the analysis of vitamin D using the 21.1 Hz frequency band, it was possible to confirm the predicted value of about 75%.

Vitamin D is one of the most studied and discussed vitamins in the field of bone and mineral metabolism diseases worldwide. The roles of various enzymes involved in vitamin D-induced hormone metabolism, their chemical structure, and their receptors have been extensively investigated. In particular, the association between body fat and vitamin D levels has been reported in a number of studies [20,21]. Vitamin D is known to act as a cofactor in the pathogenesis of obesity [2]. Those with low vitamin D levels showed significantly higher body fat percentage and waist circumference than those with sufficient vitamin D [22] and body mass index (BMI) [23]. The lower the percentage of body fat in young women, the higher the level of serum vitamin D [24]. Studies have reported that increase in vitamin D in the blood is inversely proportional to the body mass index, and that being overweight or obese decelerates the increase in vitamin D concentration [25,26]. The results of the present study showed a correlation between BMI and vitamin D and were similar to those of previous studies.

A significant correlation between vitamin D levels and systolic blood pressure [27] and a significant positive correlation between serum levels of albumin and vitamin D have been reported previously [28]. Moreover, platelet-to-lymphocyte ratio and neutrophil-to-lymphocyte ratio have also been reported to be significantly correlated with vitamin D levels, signifying the role of these parameters as markers of vitamin D levels [29]. In the present study, basophils, albumin, modification of diet in renal disease (MDRD), and LDH were confirmed to be related to the blood concentration of vitamin D.

Body fat accumulates vitamin D, inducing vitamin D deficiency in the blood, which is stronger when stratified by BMI [30]. We investigated the correlation between the concentration of vitamin D and body fat by employing impedance to measure body fat, given that impedance measurement methods that measure body fat are being used as health diagnosis tools. Based on the results obtained, we attempted to develop a method for measuring vitamin D using impedance. In this method, bioimpedance analysis employing various frequencies in the InBody device, which presented the body information, was used for vitamin D measurement. By subdividing the frequency between 20 Hz and 1 MKHz and analyzing the correlation of the impedance value measured with the blood vitamin D concentration, a frequency of 21.1 Hz was confirmed to reflect the blood vitamin D concentration at optimum levels. Impedance and vitamin D were found to maintain a positive relationship in mice [13], and similar results were confirmed in this study.

Various methods have been proposed for the non-invasive measurement of vitamin D. Methods for measuring vitamin D that do not use blood include skin color [31], body mass index [32], and hair [9]. The correlation with the amount of vitamin D in the blood was confirmed using saliva [33], and skin autofluorescence was confirmed to be related to vitamin D through an AEG reader in patients with type 2 diabetes [34]. The amount of vitamin D can be predicted by analysis using independent parameters such as age, sex, weight, height, body mass index (BMI), waist circumference, body fat, bone mass, exercise, sun exposure, and milk intake [35]. This study aims to develop a prediction algorithm through skin impedance measurement by suggesting a new alternative vitamin D measurement method. In a previous study, we confirmed that impedance measurements in mouse skin were correlated with the amount of vitamin D in the blood [13], and this study confirmed its applicability to humans. Body mass index is also similar to the method of measuring using bioelectrical signals, but this study can predict the level of vitamin D more conveniently through impedance measurement in the skin.

Vitamin D can be used in a variety of ways as an indicator of health and disease. It can be used as a biomarker to predict the clinical outcome of cutaneous melanoma [36] and as an indicator of muscle performance improvement in healthy elderly people with low physical activity [37], and its association with metabolic syndrome is known [38]. Therefore, continuous administration of vitamin D could facilitate health maintenance.

The limitation of the present study is that the number of participants in this study is small, precluding more diverse analyses. More research is needed for further analysis and is presumed to increase the reliability of the analysis algorithm by reflecting various factors such as the age and the health status of the subjects.

The present study proposed an efficient method to determine vitamin D level using a non-invasive, user-friendly method and is expected to provide an effective response in vitamin D-related diseases. In addition, this method may be utilized in a wearable device for measuring vitamin D level.

## 5. Conclusions

This study presented a novel measurement method to predict the concentration of vitamin D in the body using biosignals. The percentage of body fat is found to be correlated with the concentration of vitamin D, and we confirmed a frequency at which the concentration of vitamin D in the body can be measured by skin impedance. Experimental results confirmed that skin impedance measurements can predict the concentration of vitamin D in the body. Further research is needed to validate the reliability of the method through continuous data acquisition.

## Figures and Tables

**Figure 1 sensors-21-08118-f001:**
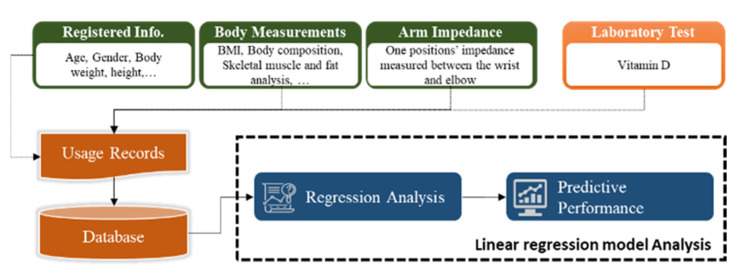
Workflow for analyzing body vitamin D levels using bioimpedance and biometric information.

**Figure 2 sensors-21-08118-f002:**
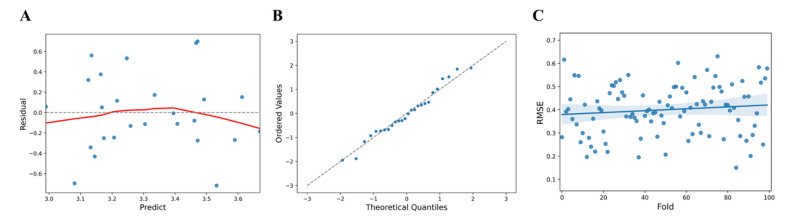
(**A**) Comparison of InBody predicted values and residuals (blue dots, standardized residual; red line, fitted line). (**B**) QQ plot of Inbody residuals (blue dots, data set; linear line, standard normal distribution). (**C**) RMSE distribution and deviation for InBody predicted values (blue dots, predicted value; blue line, Fitted values; blue area, 95% confidence limits).

**Figure 3 sensors-21-08118-f003:**
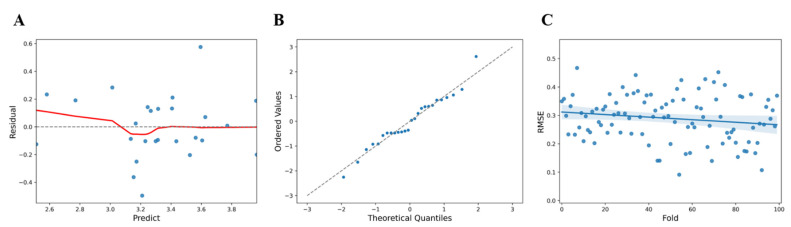
(**A**) Comparison of predicted values and residuals for health checkup parameters (blue dots, standardized residual; red line, fitted line). (**B**) QQ plot of residuals for hematology items (blue dots, data set; linear line, standard normal distribution). (**C**) RMSE distribution and deviation for health checkup parameters (blue dots, predicted value; blue line, Fitted values; blue area, 95% confidence limits).

**Figure 4 sensors-21-08118-f004:**
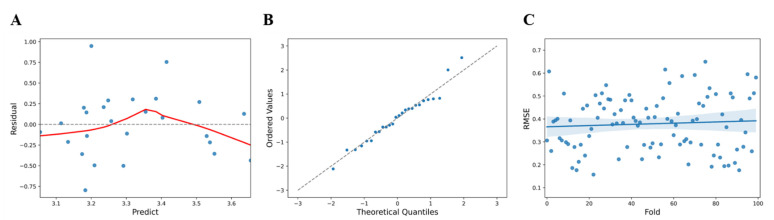
(**A**) Comparison of the predicted values and the residuals for skin impedance measurements (blue dots, standardized residual; red line, fitted line). (**B**) QQ plot of residuals for impedance measurements (blue dots, data set; linear line, standard normal distribution). (**C**) RMSE distribution and deviation for skin impedance values (blue dots, predicted value; blue line, Fitted values; blue area, 95% confidence limits).

**Figure 5 sensors-21-08118-f005:**
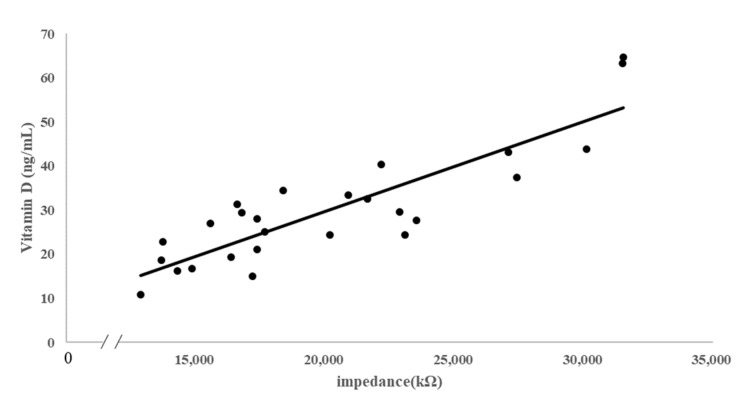
Regression analysis between skin impedance measurements and body vitamin D (dots, measured value; line, regression line).

**Table 1 sensors-21-08118-t001:** Summary of the regression model for InBody measurements.

R^2^	F-Statistics	DOF (Residuals)	DOF (Model)	*p*-Value	AIC	BIC
0.196	2.803	23	2	0.0814	27.91	31.68

DOF (Residuals): value of all observed variables minus the number of variables considered in the regression model. DOF (Model): number of variables considered in the regression model.

**Table 2 sensors-21-08118-t002:** Regression model coefficients for InBody measurements.

	Coefficient	Standardized Error	T	*p* > |t|	95% CI	VIF	RMSE
Constant	−15.1132	8.909	−1.696	0.103	[−33.543, 3.317]		0.3688
Right leg (muscle analysis by part)	3.817	1.875	2.036	0.053	[−0.062, 7.696]	2.8201	
Body fat percentage	0.0385	0.016	2.355	0.027 *	[0.005, 0.072]	2.8201	

* *p* < 0.05.

**Table 3 sensors-21-08118-t003:** Summary of the regression model for health checkup parameters.

R^2^	F-Statistics	DOF (Residuals)	DOF (Model)	*p*-Value	AIC	BIC
0.714	9.981	20	5	6.64 × 10^−5^	7.044	14.59

DOF (Residuals): value of all observed variables minus the number of variables considered in the regression model. DOF (Model): number of variables considered in the regression model.

**Table 4 sensors-21-08118-t004:** Regression model coefficients for health checkup parameters.

	Coefficient	Standardized Error	t	*p* > |t|	95% CI	VIF	RMSE
Constant	−2.4515	1.735	−1.413	0.173	[−6.07, 1.167]		0.2199
Basophils	−0.9072	0.19	−4.765	0.000 **	[−1.304, −0.51]	1.064	
Albumin	1.3068	0.317	4.116	0.001 **	[0.645, 1.969]	1.1865	
Female	0.5616	0.126	4.447	0.000 **	[0.298, 0.825]	1.6381	
MDRD	−0.0087	0.004	−2.38	0.027 *	[−0.016, −0.001]	1.5744	
LDH	0.0021	0.001	2.239	0.037 *	[0.000, 0.004]	1.1857	

* *p* < 0.05; ** *p* < 0.005.

**Table 5 sensors-21-08118-t005:** Summary of the regression model for skin impedance measurements.

R^2^	F-Statistics	DOF (Residuals)	DOF (Model)	*p*-Value	AIC	BIC
0.159	4.543	24	1	0.0435	27.07	29.59

DOF (Residuals): value of all observed variables minus the number of variables considered in the regression model. DOF (Model): number of variables considered in the regression model.

**Table 6 sensors-21-08118-t006:** Regression model coefficients for skin impedance measurements.

	Coefficient	Standardized Error	t	*p* > |t|	95% CI	VIF	RMSE
Constant	−4.2907	3.57	−1.202	0.241	[−11.659, 3.078]		0.3771
21.1 Hz	0.4419	0.207	2.131	0.043 *	[0.014, 0.87]	1	

* *p* < 0.05.

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
