# Peer review of "A Novel Biosensor and Algorithm to Predict Vitamin D Status by Measuring Skin Impedance"

_sensors, 2021, doi:10.3390/s21238118_

Round 1

Reviewer 1 Report

The authors have described how to measure Vitamin D using bioelectric impedance. This study will be helpful to non-invasively measure the Vitamin D level, which can be the root cause for many diseases and conditions. However, the study has the following concerns:

  1. Bioelectric measurements are not direct but derivative. Hence, the sample size for such studies should be large. In contrast, the sample size used here is very low. If possible, the authors should increase the sample size for this study.

  1. As the InBody measurement might have significant disturbance with different body conditions that could affect body water level, the authors should, if possible, explain their model's adaptation in such cases.

  1. Patients were chosen from only one geo-location. The authors should consider expanding their samples to other geo-location with different Sun exposures (or mention this as a future study).

  1. Some diet conditions were observed to affect the vitamin D level temporarily. Patients' diet before the impedance measurement should be discussed if possible.

See the attached document for other minor concerns.

Author Response

Thank you for your review.

  1. Bioelectric measurements are not direct but derivative. Hence, the sample size for such studies should be large. In contrast, the sample size used here is very low. If possible, the authors should increase the sample size for this study.

Response 1:

Biosignals do not directly measure vitamin D. As mentioned, this study is a method of measuring vitamin D in an indirect way. In fact, the sample of 26 people in this study is very small. This research team has previously confirmed the relationship between vitamin D and skin impedance using a mouse model (ref. 14). As an extension of this study, this study hypothesized that there would be a way to check the level of vitamin D in humans in a similar way. As mentioned, the reliability is somewhat lower due to the lack of samples, but so far, no studies have clinically confirmed the relationship between skin impedance and vitamin D. Through this study, although the reliability is rather low, we were able to confirm the pattern of relationship between vitamin D and skin impedance. We are planning follow-up studies to secure more reliable data in the future, and we will use more samples to secure reliable data.

  1. As the InBody measurement might have significant disturbance with different body conditions that could affect body water level, the authors should, if possible, explain their model's adaptation in such cases.

Response 2:

That's a good point. In-Body measurement status may appear differently depending on the time of measurement and physical condition. On the one hand, the InBody measurement value does not change significantly unless there is a sudden change in body condition. It has already been reported that in-body measurements in existing studies are not perfect, but have considerable reliability. In order to secure the reliability of the study, the tester performed the InBody measurement on the same day as the blood test, and participated in the measurement under conditions equivalent to the health checkup.

The following has been added.

Line 109-111.

“Participants performed InBody measurements on the same day as the health checkup, and three times average value for analysis.”

  1. Patients were chosen from only one geo-location. The authors should consider expanding their samples to other geo-location with different Sun exposures (or mention this as a future study).

Response 3:

In the study using InBody, all participants were measured indoors under the same conditions. If the environment (indoor or outdoor) changes, the results may differ as mentioned. It has been reported that vitamin D is produced by exposure to sunlight, so it is possible to predict the increase or decrease of vitamin D with the degree of exposure to sunlight. Related research on this will be carried out in the future. However, it was predicted that the concentration of vitamin D in the body would not change rapidly, and this research team was also concerned about this. We reviewed the various factors that determine the amount of vitamin D in the body. In this study, it was judged that it was appropriate to use the participants' daily status as a standard. In the future, we will provide more accurate information by adding users' lifestyles such as geographic factors and exposure to sunlight. 

  1. Some diet conditions were observed to affect the vitamin D level temporarily. Patients' diet before the impedance measurement should be discussed if possible.

Response 4:

That's a good comment. It is agreed that the patient's dietary status may affect the concentration of vitamin D in the body. Various factors affect the concentration of vitamin D, and statistical analysis of these factors is required. However, if the factors mentioned above are included, it is necessary to discuss the difficulty of analysis and whether it can be used universally. The purpose of this study was to develop an algorithm that can easily predict the amount of vitamin D, and in the future, factors that can reflect various environmental factors will be reflected in the algorithm.

  1. See the attached document for other minor concerns.

Response 5:

The mentioned part has been corrected.

Thank you very much for reviewing our paper.

It has been very helpful in developing a more reliable predictive algorithm for vitamin D in the future.

This study has contributed greatly to future research including the paper.

Thank you.

Reviewer 2 Report

In the current article, the authors are developing Vitamin D sensors based on impedance data and algorithms.  The article can be accepted after minor revision.

Here I am mentioning my concerns.

  1. Abstract looks very general authors need to write abstracts precisely that starting sentences.
  2. Authors should discuss previous literature which deals with the algorithm for Vitamin D sense.
  3. In the article, authors should give more details of their algorithms (the details given in lines number 136-139 are not sufficient)
  4. The quality of figure 2 should be improved
  5. Authors should give more details on Q-Q models
  6. The quality of figure 4 is very low
  7. How authors are validating the algorithm results has to be explained in a detailed manner.
  8. The author has to mention how many times he has repeated the experiments to confirm the results.
  9. Image quality is very poor, the author has to draw axis properly with software like GNUPLOT or ORIGIN.
  10. The materials section should be present in a more crisp form.
  11. The discussion is very poor.
  12. Some other quantitative and qualitative values should be added in the abstract.

Abstract need to be revised

  1. Authors have to take care of grammar issue

Author Response

Thank you for your review.

  1. Abstract looks very general authors need to write abstracts precisely that starting sentences.

Response 1:

Abstract has been partially modified.

  1. Authors should discuss previous literature which deals with the algorithm for Vitamin D sense.

Response 2:

I have inserted the part you mentioned.

Line 52-60.

“The vitamin D test is an accurate way to measure the amount of vitamin D in the body. Indirectly, vitamin D is a measurement method using hair [10], and a Insufficiency Prediction score that reflects individual characteristics using a multivariate logistic regression model for overweight, physical activity, winter season, sun exposure, etc.[11]. The amount of vitamin D in mouse skin was measured using an impedance-based quantum analyzer as a non-invasive method using biosignals[12], and there is a report that body mass index, body weight, and body fat are related to the amount of vitamin D in the body[13, 14]. However, there was no study that predicted the level of vitamin D by measuring skin impedance in humans.”

  1. In the article, authors should give more details of their algorithms (the details given in lines number 136-139 are not sufficient)

Response 3:

Relevant content has been inserted.

Line 155-162.

“For data analysis elements, data collected from the participant's gender, age, and measurement device were used, and the correlation with vitamin D was analyzed, and measurement parameters that could be used to estimate the amount of vitamin D were selected. In this study, data from 26 people were used for analysis, and in order to improve the performance evaluation of the dataset, cross-validation and error indicators were converted into units similar to the actual values for analysis. The analysis values were used as statistical graphics by using a visualization library.”

  1. The quality of figure 2 should be improved

Response 4:

The figure has been modified.

  1. Authors should give more details on Q-Q models

Response 5:

I have added and edited the content.

Line 222-224.

“Using QQ plot (Quantile-Quantile Plot), it was reviewed whether the assumption of normal distribution was appropriate.”

  1. The quality of figure 4 is very low

Response 6:

The figure has been modified.

  1. How authors are validating the algorithm results has to be explained in a detailed manner.

Response 7:

It has been modified by adding related content.

Line 324-332.

“We tried to develop an algorithm to predict the concentration of vitamin D in the body based on skin impedance. In order to select a frequency at which impedance can be measured, the 20 Hz–1 MHz frequency band was divided into 400 sections on the participant's skin to check the impedance. The data obtained from InBody confirmed the correlation between body composition content and skin impedance through linkage analysis. To predict the level of vitamin D in the body using impedance, the frequency band of 21.1 Hz was confirmed through statistical analysis using the measured impedance value. As a result of confirming the predictive value for the analysis of vitamin D using the 21.1 Hz frequency band, it was possible to confirm the predicted value of about 75%.”

  1. The author has to mention how many times he has repeated the experiments to confirm the results.

Response 8:

InBody and impedance measurements were performed three times and the average value was used for analysis.

Related content has been added to the method.

Line 108-111.

“Participants performed InBody measurements on the same day as the health checkup, and three times average value for analysis.”

  1. Image quality is very poor, the author has to draw axis properly with software like GNUPLOT or ORIGIN.

Response 9:

The figure has been modified.

  1. The materials section should be present in a more crisp form.

Response 10:

A data analysis part and a part about participants have been added.

Line 155-162.

“For data analysis elements, data collected from the participant's gender, age, and measurement device were used, and the correlation with vitamin D was analyzed, and measurement parameters that could be used to estimate the amount of vitamin D were selected. In this study, data from 26 people were used for analysis, and in order to improve the performance evaluation of the dataset, cross-validation and error indicators were converted into units similar to the actual values for analysis. The analysis values were used as statistical graphics by using a visualization library.”

  1. The discussion is very poor.

Response 11:

Related content has been added.

Line 366-379.

“Various methods have been proposed for non-invasive measurement of vitamin D. Methods for measuring vitamin D that do not use blood include skin color[33], body mass index[34], and hair[12]. The correlation with the amount of vitamin D in the blood was confirmed using saliva[35], and skin autofluorescence was confirmed to be related to vitamin D through an AEG reader in patients with type 2 diabetes[36]. The amount of vitamin D can be predicted by analysis using independent parameters such as age, sex, weight, height, body mass index (BMI), waist circumference, body fat, bone mass, exercise, sun exposure and milk intake [37]. This study aims to develop a prediction algorithm through skin impedance measurement by suggesting a new alternative to the vitamin D measurement method. In a previous study, we confirmed that impedance measurements in mouse skin were correlated with the amount of vitamin D in the blood [14], and this study confirmed its applicability to humans. Body mass index is also similar to the method of measuring using bioelectrical signals, but this study can predict the level of vitamin D more conveniently through impedance measurement in the skin.”

  1. Some other quantitative and qualitative values should be added in the abstract.

Response 12:

Related content has been modified.

Line 24-25.

“An impedance measurement frequency of 21.1 Hz reflects the blood vitamin D concentration at optimum levels, and about 75% of the confidence level for vitamin D in the body was confirmed.”

  1. Abstract need to be revised

Response 13:

Abstract has been partially modified.

  1. Authors have to take care of grammar issue

Response 14:

English proofreading was performed in Editage (www.editage.co.kr).

Thank you very much for reviewing our paper.

It has been very helpful in developing a more reliable predictive algorithm for vitamin D in the future.

This study has contributed greatly to future research including the paper.

Thank you.

Reviewer 3 Report

This work combined machine learning and a skin impedance sensor for detecting Vitamin D in humans. The highlights are not summarized very well, and the introduction may need to be supplemented. Language needs to be polished, and I recommend it can be accepted after minor revision.

  1. Many researchers have already used the machine learning-assisted analyte quantization method. What is the pioneering point of this work?
  2. The R2 value for the regression equation is still very low based on skin impedance measurements. How to explain this result?
  3. The data were collected only from 26 participants. How does such a small sample meet the accuracy of the experiment?
  4. Since machine learning is a highlight for this work, some background and advantages of ML for biosensing should be supplied in the introduction.
  5. The meaning of parameters should be explained, like p-value, T, F-statistics, etc.
  6. The legend of figures should be given. Like in Figure 2, it should add legends for blue dots, red line, and blue line, etc.
  7. In the introduction section, even though the authors mentioned various detection methods of vitamin D, some important sensing methods are missed, like molecular imprinting technology, ELISA, etc. Some related researches are attached as references. (Sci. Rep. 7.1 (2017): 1-8.; RSC adv. 6.38 (2016): 31906-31914.; Biosens. Bioelectron. 149 (2020): 111830.; Bone Miner. Res. 31.6 (2016): 1128-1136. )
  8. English still needs to be polished. Some grammar issues and mistakes are found.
  9. The “Bio Sensor” in the title is better to write as “biosensor”.
  10. In Line 14, The “Vitamin D deficiency and excess” is better to write as “The deficiency and excess of Vitamin D”.
  11. The abstract writing should use present tense, so some descriptions like “were analyzed” “was developed” “reflected” in the abstract should be revised.
  12. Some Korean languages are found in Table 6.

Author Response

Thank you for your review.

  1. Many researchers have already used the machine learning-assisted analyte quantization method. What is the pioneering point of this work?

Response 1:

There are various methods of analyzing vitamin D through machine learning based on biometric information. The concentration of vitamin D can be predicted by complexly analyzing various factors such as skin color, body mass index, and impedance. Compared with existing methods, this study can determine the level of vitamin D only by measuring the impedance of the skin. This can have many advantages in terms of size, measurement area, and simplification of analysis algorithms in future development of measurement devices.

The following has been added.

Line 366-379.

“Various methods have been proposed for non-invasive measurement of vitamin D. Methods for measuring vitamin D that do not use blood include skin color[33], body mass index[34], and hair[12]. The correlation with the amount of vitamin D in the blood was confirmed using saliva[35], and skin autofluorescence was confirmed to be related to vitamin D through an AEG reader in patients with type 2 diabetes[36]. The amount of vitamin D can be predicted by analysis using independent parameters such as age, sex, weight, height, body mass index (BMI), waist circumference, body fat, bone mass, exercise, sun exposure and milk intake [37]. This study aims to develop a prediction algorithm through skin impedance measurement by suggesting a new alternative to the vitamin D measurement method. In a previous study, we confirmed that impedance measurements in mouse skin were correlated with the amount of vitamin D in the blood [14], and this study confirmed its applicability to humans. Body mass index is also similar to the method of measuring using bioelectrical signals, but this study can predict the level of vitamin D more conveniently through impedance measurement in the skin.”

  1. The R2 value for the regression equation is still very low based on skin impedance measurements. How to explain this result?

Response 2:

That's a good point. This study applied an algorithm to predict the level of vitamin D through skin impedance measurement indirectly. There are many different factors that determine the biosignal (skin impedance). It is determined by a number of factors, such as the condition of the skin and the fat content, and it is measured differently for each person. Skin impedance does not reflect only the level of vitamin D, and although it is currently unknown, impedance measurements are thought to have various meanings. Therefore, we first selected the frequency band that can best express the vitamin D area, and we tried to predict the level of vitamin D using this. In this study, we confirmed the pattern that skin impedance is related to vitamin D concentration. In the future, factors that can further secure reliability (sunshine, lifestyle, eating habits, etc.) will be reflected and introduced into the analysis algorithm.

  1. The data were collected only from 26 participants. How does such a small sample meet the accuracy of the experiment?

Response 3:

Biosignals do not directly measure vitamin D. As mentioned, this study is a method of measuring vitamin D in an indirect way. In fact, the sample of 26 people in this study is very small. This research team has previously confirmed the relationship between vitamin D and skin impedance using a mouse model (ref. 14). As an extension of this study, this study hypothesized that there would be a way to check the level of vitamin D in humans in a similar way. As mentioned, the reliability is somewhat lower due to the lack of samples, but so far, no studies have clinically confirmed the relationship between skin impedance and vitamin D. Through this study, although the reliability is rather low, we were able to confirm the pattern of relationship between vitamin D and skin impedance. We are planning follow-up studies to secure more reliable data in the future, and we will use more samples to secure reliable data.

  1. Since machine learning is a highlight for this work, some background and advantages of ML for biosensing should be supplied in the introduction.

Response 4:

The relevant part has been added.

Line 61-67.

“As a branch of artificial intelligence, machine learning has made remarkable progress. Deep learning remains relatively elusive in the biosensor community, but based on the analysis of sensed data, and new analysis algorithms can be created [17]. It is used for analysis of electrochemical, spectral, fluorescence biosensors, etc. In addition, biosensor data can be analyzed from various perspectives using fusion data. The connection between machine learning and biosensors will expand significantly as tools for detection, analysis and diagnostics [18].”

  1. The meaning of parameters should be explained, like p-value, T, F-statistics, etc.

Response 5:

The relevant part has been added.

Line 212-216.

“Upon data analysis, the F statistic, the ratio of two quantities that are expected to be equal under the null hypothesis, of the regression model showed a level of 2.803 at p (probability value)=0.0814, R2 =0.196 for the regression formula which showed an explanatory power of 19.6%, AIC (Akaike Information Criterion)=27.91 and BIC (Bayesian Information Criterion)=31.68”

  1. The legend of figures should be given. Like in Figure 2, it should add legends for blue dots, red line, and blue line, etc.

Response 6:

Add the blue dots, red line, and blue line names.

  1. In the introduction section, even though the authors mentioned various detection methods of vitamin D, some important sensing methods are missed, like molecular imprinting technology, ELISA, etc. Some related researches are attached as references. (Sci. Rep. 7.1 (2017): 1-8.; RSC adv. 6.38 (2016): 31906-31914.; Biosens. Bioelectron. 149 (2020): 111830.; Bone Miner. Res. 31.6 (2016): 1128-1136. )

Response 7:

Two papers were listed as references.

Ref. 10, 11

  1. English still needs to be polished. Some grammar issues and mistakes are found.

Response 8:

English proofreading was performed in Editage (www.editage.co.kr).

  1. The “Bio Sensor” in the title is better to write as “biosensor”.

Response 9:

It has been corrected.

  1. In Line 14, The “Vitamin D deficiency and excess” is better to write as “The deficiency and excess of Vitamin D”.

Response 10:

It has been corrected.

  1. The abstract writing should use present tense, so some descriptions like “were analyzed” “was developed” “reflected” in the abstract should be revised.

Response 11:

It has been corrected.

  1. Some Korean languages are found in Table 6.

Response 12:

It has been corrected.

Thank you very much for reviewing our paper.

It has been very helpful in developing a more reliable predictive algorithm for vitamin D in the future.

This study has contributed greatly to future research including the paper.

Thank you.
